# Biophysical Profiling of Sickle Cell Disease Using Holographic Cytometry and Deep Learning

**DOI:** 10.3390/ijms241511885

**Published:** 2023-07-25

**Authors:** Cindy X. Chen, George T. Funkenbusch, Adam Wax

**Affiliations:** BIOS Lab, Department of Biomedical Engineering, Duke University, Durham, NC 27708, USAa.wax@duke.edu (A.W.)

**Keywords:** sickle cell disease, flow cytometry, holography, quantitative phase imaging, deep learning, detection, monitoring

## Abstract

Sickle cell disease (SCD) is an inherited hematological disorder associated with high mortality rates, particularly in sub-Saharan Africa. SCD arises due to the polymerization of sickle hemoglobin, which reduces flexibility of red blood cells (RBCs), causing blood vessel occlusion and leading to severe morbidity and early mortality rates if untreated. While sickle solubility tests are available to sub-Saharan African population as a means for detecting sickle hemoglobin (HbS), the test falls short in assessing the severity of the disease and visualizing the degree of cellular deformation. Here, we propose use of holographic cytometry (HC), a high throughput, label-free imaging modality, for comprehensive morphological profiling of RBCs as a means to detect SCD. For this study, more than 2.5 million single-cell holographic images from normal and SCD patient samples were collected using the HC system. We have developed an approach for specially defining training data to improve machine learning classification. Here, we demonstrate the deep learning classifier developed using this approach can produce highly accurate classification, even on unknown patient samples.

## 1. Introduction

Sickle cell disease (SCD) is an inherited hemoglobin disorder condition most commonly found in sub-Saharan Africa, India, the Mediterranean and the Middle East population [1]. Characteristics of the disease include chronic hemolytic anemia, acute pain, organ damage and significantly shorter lifespans [2]. The pathophysiology of SCD mainly involves the rapid polymerization of hemoglobin (HbS) after deoxygenation, causing deformation of red blood cell (RBC) morphology and disrupting blood circulation [3]. Consequently, vaso-occlusion occurs due to the inflexibility and highly adhesive nature of sickled RBCs [3]. Organ failures are common complications of SCD from repeated vaso-occlusive processes and often lead to increased mortality [4,5].

High mortality rates are particularly prevalent in the under-five SCD age group, constituting 50–90% of newborns in sub-Saharan Africa with the β-globin S gene mutation [6]. In North America, mortality rates for SCD-related complications have nearly been eliminated due to the availability of comprehensive blood tests and newborn screening protocols [6,7]. However, access to specialized SCD diagnosis laboratory equipment such as high-performance liquid chromatography or electrophoresis is extremely limited in low-resource countries, and thus, a sickle solubility test is the most affordable and available testing method [8]. The test looks for the presence of deoxygenated HbS precipitate in phosphate buffer, formed from reactions induced by sodium bisulfite. While a ‘positive’ test result confirms the presence of HbS, it is incapable of providing detailed information on the percentage of sickling cells in a heterogeneous blood sample and is not effective for HbS levels lower than 15–20% [9]. More accurate and accessible diagnostic tools are needed for SCD management.

Healthy and sickle cell trait (SCT) individuals generally have normal hemoglobin levels, normal RBC shape, and have no significant clinical or hematologic manifestations [10,11]. The defining features of SCD on a microscopic level is the gradual morphological deformation of select sickle cell erythrocytes due to the progressively increasing polymerization of HbS under low oxygen tension conditions [12]. Morphological and biomechanical profiling of single sickle cells has been accomplished using quantitative phase microscopy (QPM), a powerful label-free imaging modality [13,14]. Yet, the application of QPM for cytological diagnosis has been limited by low imaging cell counts. Recent advances in QPM have led to higher imaging cell counts in an effort to provide a more complete view on overall cell population morphology and, thus, bring more diagnostic value. For example, high-throughput QPM is achieved via the fast-scanning time-stretch approach, producing a state-of-the-art throughput of 77,000 cells/second [15]. However, these results are achieved at the expense of instrument complexity, large amounts of computing power and lengthy image reconstruction times. As an alternative, we recently introduced holographic cytometry (HC) as a simple low-cost, high-throughput QPM system, based on an off-axis Mach–Zehnder configuration that only requires phase reconstruction times in the order of milliseconds [16,17].

In this study, we seek to evaluate HC’s clinical applicability to SCD screening and monitoring. Millions of single cell images extracted from just a few drops of blood are imaged using HC in a few minutes and passed through a machine learning algorithm for detailed diagnosis. Here, we advance the algorithmic approach to improve identification of cells from SCD patients. While cells from a given SCD patient sample can exhibit a wide range in degree of cellular deformation [18,19] due to the heterogeneity in SCD cellular morphology, here, we define a set of selection criteria to refine a training dataset that only contain extremely sickled cells. Upon using this filtered dataset for training, we developed deep learning algorithms that can accurately predict the percentage of severely sickled cells in unknown patient samples.

## 2. Results

Example box plots and histograms are shown in Figure 1 for two selected morphological parameters. Overall, the trends show that the SCD samples have a higher average eccentricity ratio and slightly lower average mean phase than normal samples. The histograms show that there is great overlap in eccentricity and mean phase between the two different cell types even though there is a statistically significant difference (*p* < 0.001) at the population level. It is observed that a small proportion of the morphological parameter distributions for the SCD population do not overlap with the normal dataset. A dataset refinement selection criterion is developed based on the differences found in this population distribution mismatch. We implement a selection criterion that automatically generates an accurately labeled training dataset (error rate less than 5%). For the training of the machine learning models below, we present a side-by-side comparison between models trained by the refined and unrefined datasets.

### 2.1. Logistic Regression Training and Testing

Initially, the logistic regression (LR) model was trained on the entire dataset of morphological parameters. All SCD sample data and all normal sample data were grouped and randomly scrambled forming two different data pools. Initially, the model was trained using 10,000 SCD morphological parameters and 10,000 normal morphological parameters randomly selected from the data pools. In Table 1, the logistic regression-all (LR-ALL) model presents the training performance of this approach, achieving ~76% accuracy (77% sensitivity and 75% specificity). In a second approach, a smaller training dataset containing only data meeting the selection criteria described in Section 3.5 is used to train the LR model. In Table 1, the logistic regression-selected (LR-SEL) model presents the training performance of this approach, achieving 99.28% accuracy (99.53% sensitivity and 99.03% specificity). To calculate these performance metrics, 90% of the input data was used for training the model, while 10% of the data was used for validation. In comparison, the LR-SEL model significantly outperformed LR-ALL model in training sensitivity, specificity and accuracy.

To further evaluate the models’ performance, we constructed a test dataset consisting of archived normal RBC data from a previous study [16] and one additional SCD patient sample that was not included in the training. The testing dataset contains more than 1.6 million cells’ morphological parameters, which are passed through both LR-SEL and LR-ALL, with the results shown in Table 2 and Table 3. While normal subjects accuracy levels for LR-ALL ranged from 33.33% (sample D) to 99.64% (sample A), LR-SEL exhibited higher accuracy ranging from 83.77% (sample B) to 98.66% (sample A). The LR-ALL model’s accuracy level for the one SCD patient is indistinguishable from normal subjects; unfortunately, the model predicted sample D (66.67%) and sample E (43.25%) to have higher diseased cell counts than sample 3 (40.84%). In comparison, LR-SEL predicts sample 3 to have significantly higher diseased cell counts than all healthy subject samples, and all healthy subject samples have >83% predictions of normal, as shown in Table 3. Overall, the LR-SEL model presents an average accuracy of 93.17% across samples A–E, where four out of the four normal subjects samples are >94%.

### 2.2. Convolutional Neural Network Training and Testing

Table 4 summarizes the convolutional neural network (CNN) testing accuracy results, including convolutional neural network-all (CNN-ALL) model trained on the entire body of cell images in the unrefined SCD and unrefined normal datasets. A total of 100,000 SCD images and 100,000 normal images were randomly selected from the data pool for the training of CNN-ALL. Similarly to LR-ALL, while the CNN-ALL model showed great success in correctly predicting normal samples (92–95%) for samples A to C, it performed poorly for samples D, E and 3 (48–53%). In comparison, switching to the use of a more selective dataset greatly improves performance. Table 5 summarizes the testing accuracy of the convolutional neural network-selected (CNN-SEL) model trained by the refined SCD and refined normal dataset. CNN-SEL achieved high accuracy levels for all normal samples (92–99%). Only less than 8% of cell images from samples A to E were mistakenly counted as diseased, clearly distinct from sample 3′s diseased cell count of 25.96%. This model predicts a slightly higher prediction of normal cells for the SCD sample from sample 3 but still remains distinct from the other samples.

We observe that the dataset refinement process greatly improved the diagnostic ability of the models when analyzed using a receiver operator curve (ROC). The area under curve (AUC) increased from 0.84026 (Figure 2A) for LR-ALL to 0.99897 (Figure 2B) for LR-SEL. Similarly, we report an increase in AUC from 0.9657 (Figure 3A) for CNN-ALL to 0.9996 (Figure 3B) for CNN-SEL. The switch in training dataset to a more selective SCD subset has significantly increased the models’ class separation capacity.

## 3. Materials and Methods

### 3.1. System

The experimental setup consists of the HC imaging system [20] accompanied by artificial intelligence algorithms to realize SCD diagnosis. Shown in Figure 4, the imaging system is a Mach–Zehnder interferometer that consists of pathlength-matched sample and reference arms, where images of RBCs flowing within microfluidic channels are captured in the form of single-cell holograms. The overall magnification of the system is 33×, and the field of view covers 16 channels at once. The camera (Dalsa HS-40-04K40-00-R) acquires 300 frames per second synchronized to an acousto-optic modulator that pulses a 640 nm wavelength laser (PicoQuant Fast Switched Diode Laser FSL 500). The 350μs pulses prevent streaking effects by minimizing the blurring due to motion of the flowing cells.

### 3.2. Image Reconstruction

Single-cell phase images are reconstructed from interferograms using Fourier transform, phase unwrapping, digital refocusing and segmentation algorithms, at rates up to 150 ms/cell [20]. Twenty-five morphological parameters are extracted from each single-cell image. Customized exclusion parameters are used to eliminate images of cell clumps and debris from the dataset. Normal cell images with mean phase values below 0.4 rad and SCD cell images with mean phase values below 0.3 rad are excluded from the final dataset. In total, 1981 SCD images and 5552 healthy images were excluded from our analysis. 

### 3.3. Microfluidics

Customized lithographic patterns are fabricated onto blank Si wafer disks through SU-8 etching process. The finished Si mold is used to form microfluidic channels (Figure 4) using polydimethlsiloxane (PDMS), mixed at a 10:1 ratio with PDMS curing agent. The mixture is baked in an oven at 85 °C for two hours, allowing the channels to cure and solidify. Subsequently, the cured PDMS slabs are then plasma bonded to glass coverslips in a reactive ion etcher chamber. Entry and exit ports to the channels are punctured onto PDMS slab prior to bonding. During each data acquisition session, flow rate is set to 10 μL/min.

### 3.4. Blood Samples

Fresh packed RBCs (pRBCs) from 2 healthy donors and 3 SCD donors (Table 6) were purchased from an external vendor (BioIVT) for the purpose of this study. An amount of 50 μL of pRBCs was suspended in 5 mL of 20% bovine serum albumin (BSA) solution and pumped into the microfluidic channel using a syringe pump. Archived RBC HC imaging data (Table 7) from our 2021 study, which were analyzed here, were also processed under the same protocol [17]. 

### 3.5. Selective Search and Training Set Refinement

In order to construct an automatically labelled ground truth training set, with minimal errors, a search criterion has been established to define features which differ at a ratio of 21:1 for the percentage of SCD population that shows this characteristic versus percentage of normal population above/below the searched threshold value. Previous studies have shown that the logistic regression achieves the highest training accuracy levels when trained with datasets with minimum size of 5000 images [17]. The cutoff ratio is set based on finding the balance point where there is a high selectivity yet maintains a reasonable diseased cell count (at least 5000 cells) in the final combined dataset. We term these tails as critically sickled cells. The final training set consists of sample 1 and sample 2 data that are filtered using the search criteria. 

For each morphological parameter, a population-wide search is conducted to find the histogram tail threshold where the search criteria can be satisfied (Figure 5). As an example, 0.8118 radians is the threshold calculated for the maximum phase histograms. For every 21 sickle cells that are below this maximum phase threshold, only 1 healthy cell will exist in the same regime. Several of the morphological parameters, including max phase, standard deviation of phase, top 25% optical path length, max phase gradient, eccentricity and elongation ratio all had tail distributions that satisfied the search criterion. These tail subsets are grouped together, and the union of the sets yielded 18,925 SCD cell images for training. Another 18,925 normal cell images were randomly selected from Sample 4 and Sample 5 to add to the training set. Test datasets consist of all archived data shown in Table 7 and Sample 3 SCD data. Figure 5 presents a graphical overview of the tail selection process.

### 3.6. Logistic Regression and Deep Learning

Two different training sets were used for constructing two different variants, each of LR and CNN models. LR-ALL and CNN-ALL models were trained using unrefined data from SCD and normal samples, whereas LR-SEL and CNN-SEL models were trained using refined data from the above-described selective search process. 

Morphological parameters extracted from samples 1, 2, 4 and 5 were used for training the LR algorithms to distinguish between SCD and normal. The testing dataset consisted of morphological parameters extracted from samples A–E and sample 3. In total, the LR-ALL model was trained by an unrefined dataset of 2 × 100,000 × 25 parameters (#classes × #cells × #parameters), and the LR-SEL model was trained by a refined dataset of 2 × 18,925 × 25 parameters.

Rather than first extracting morphological parameters, single-cell images from samples 1, 2, 4 and 5 were used for training the deep learning neural networks. Single-cell images from samples A–E and sample 3 were used for testing the deep learning neural network’s performance. Overall, CNN-ALL was trained on an unrefined dataset of 2 × 100,000 (#classes × #cells) cell images, and CNN-SEL was trained on a refined dataset of 2 × 18,925 cell images to make inferences on 6 unknown patient samples.

## 4. Discussion

We observe that the sickle sample and normal sample morphological parameters histograms are mostly overlapped; however, a nontrivial number of the sickle cells in the distribution exhibit nearly no phenotypic similarities to healthy cells. This unique subset of sickle cells can be extracted from the distribution through implementing a 21:1 SCD-to-normal ratio thresholding criterion. As described above, this means that each criterion is defined by the region in the histogram where the population of SCD cells have 21× greater incidence of that chosen parameter than the normal cell population. The extraction of morphological parameters that uniquely identify sickling cells is a necessary step to constructing a meaningful, pure SCD ground truth set. Without thresholding, the large overlap between the morphological parameters of sickle and normal cells sample will confound the discrimination capacity of a classification algorithm. Furthermore, through refining the training set, we have narrowed down the focus of the classification algorithm to specifically differentiate SCD cells from healthy cells and greatly increase accuracy. The near ideal ROC curves (Figure 2B and Figure 3B) for LR and CNN algorithms when trained on the refined data, indicates that the algorithm practically predicted no false positives and would excel at identifying critically sickled cells. Overall, we have developed a group of metrics that can delineate critically sickling cells in a heterogeneous SCD sample.

When training is switched from the unrefined to refined dataset, significant improvements in sensitivity, specificity and accuracy are observed in the machine learning models’ performance. The ROC graph shown for refined dataset (Figure 2B and Figure 3B) has a greater AUC than the unrefined dataset (Figure 2A and Figure 3A), indicating that enforcement of the selection criterion greatly improved the models’ class separation capacity between positive and negative class points. Between LR and CNN, CNN-SEL’s AUC (0.9996) is greater than LR-SEL’s AUC (0.99897), and CNN-SEL has an overall higher average normal sample accuracy level than LR-SEL (96.71% > 93.17%); thus, it is evident that CNN performed better at classifying the SCD class. Implementation of the search criterion is critical for the realization of an accurate depiction of a heterogeneous cytological blood sample for clinical settings.

While it is possible to produce classification decisions by simply evaluating the absolute number of critically deformed cells in any unknown patient sample, it would be difficult to analyze a sample with a large number of cells with this approach in real-time clinical settings. Out of the 471,792 sickle cells analyzed, only 18,925 cells matched our critically sickled cell criterion, equating to less than 5% of overall population. To ensure that quick, accurate classification decisions can be provided in clinical care settings, where a priority is placed on receiving quick, actionable results, it would be likely that smaller sample sizes such as ~5000 cells are used for analysis. Under such circumstances, deep learning models are superior to simply using a threshold to analyze a large population dataset approach because they can provide rapid decisions with much smaller sample sizes. Despite the fact that regular thresholding deemed that less than 5% of the population is critically sickled, deep learning found more than 25% of the cell population had sufficient differences to be identified as diseased. Deep learning potentially has the capacity to make a more nuanced decision than regular thresholding. Cells exhibiting slightly less severe yet still sickled deformation can be captured by the deep learning model, while traditional thresholding would fail to do so.

Both LR and CNN produced extremely high accuracy rates in identifying healthy subject samples yet lower counts of critically sickled cells in a SCD patient sample. Although only 25–29% of SCD patient cells were identified as SCD, this is within the expectation that not all RBCs in a given sample may be critically sickled and that only a portion of the RBCs are sufficiently altered to appear distinct from healthy RBCs. Since the deep learning network showed the worst performance on Sample D, resulting in 92.62% accuracy, we can suggest the prediction of 8% diseased cell count as the lower bound and 25% as the upper bound for a decision model. For patient samples which the deep learning model inferred to have more than 8% critically sickled cells, the decision model would recognize the entire sample as SCD. Potentially, the sample-to-sample variation in percentages (8–25%) could be used as a metric to evaluate the overall disease severity of a given patient and possibly provide information that could be used to predict sickling crisis and response to therapeutic treatments. 

While SCD individuals are homozygous for HbS, SCT has a heterozygous genotype and is often considered a benign condition, with morbidity and mortality rates similar to the general population [21]. We predict that the diseased cell count for SCT individuals would present at less than 8%. Future work should be focused on incorporating SCT data into the selective training algorithm. A side-by-side comparison of morphological parameter histograms may aid in the development of a set of additional selection criteria for differentiating SCT individuals from healthy subjects and SCD patients. At the other end of the spectrum, we would predict that some SCD individuals will have diseased cell counts above 25%. Patients with different SCD severity may exhibit different levels of diseased cell count. Additional SCD patient data may help fill the upper gaps in the current decision model. 

One limitation of our approach is that other types of morphology-altering blood diseases have not been considered in this study. Previous studies have demonstrated QPM as a useful tool for detecting RBC anomalies such as changes in sphericity due to storage and water content changes due to mechanical compression [22,23]. The selective training methods presented in this paper can further enhance QPM’s sensitivity to RBC morphology changes. With further model development, the HC modality may potentially be used to detect blood disorders that cause RBC deformation, such as hereditary spherocytosis and beta thalassemia. 

## 5. Conclusions

We present our findings in utilizing HC as a high-throughput screening tool for SCD based on analysis of single RBC images. Two models, one based on logistic regression and one based on deep learning, have been developed and shown capable of distinguishing normal patient samples from SCD patient samples. We also report a method to identify critically sickled cells in SCD samples, which shows potential for being developed into a metric for disease severity assessment. Future work should consider expanding the current RBC data library to including sickle cell trait and other types of morphology-altering blood diseases.

## Figures and Tables

**Figure 1 ijms-24-11885-f001:**
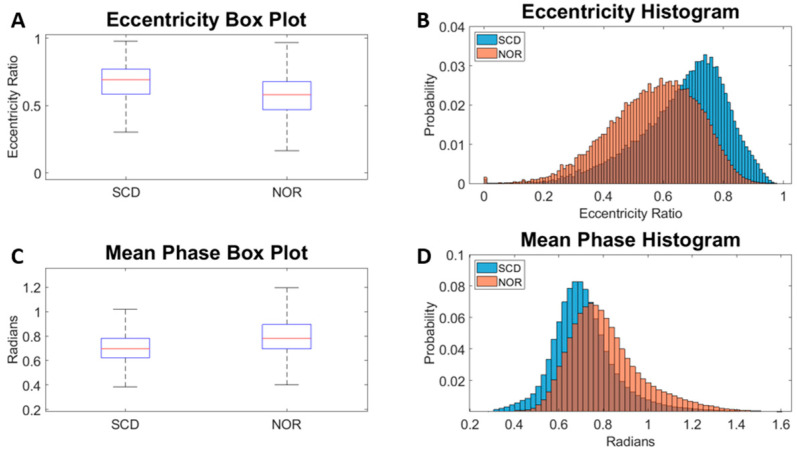
Example morphological parameters box plots and histograms. *p*-values < 0.001 for parameters shown here. Eccentricity refers to the ratio of the distance between the foci of the ellipse and its major axis length. Mean phase refers to the mean of the phase values obtained from single cell phase image. SCD: samples from sickle cell disease patients. NOR: samples from normal subjects.

**Figure 2 ijms-24-11885-f002:**
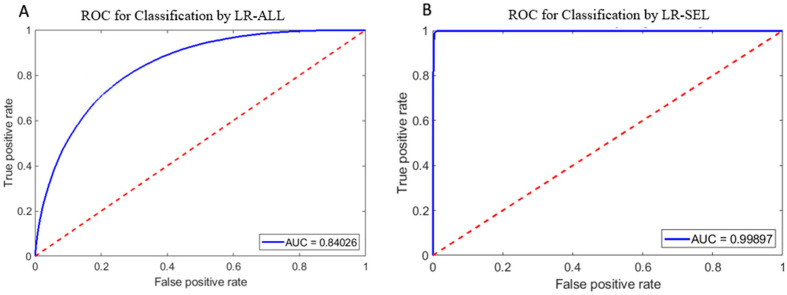
Blue: Receiver operating curve (ROC) for classifier trained with unrefined dataset. Red dash: ROC for random classifier. (**A**). ROC for LR-ALL. (**B**). ROC for LR-SEL. AUC: area under curve.

**Figure 3 ijms-24-11885-f003:**
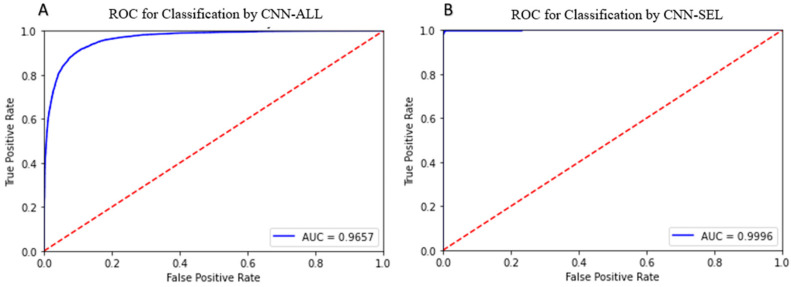
Blue: ROC for classifier trained with refined dataset. Red dash: ROC for random classifier. (**A**). ROC for CNN-ALL. (**B**). ROC for CNN-SEL. AUC: area under curve.

**Figure 4 ijms-24-11885-f004:**
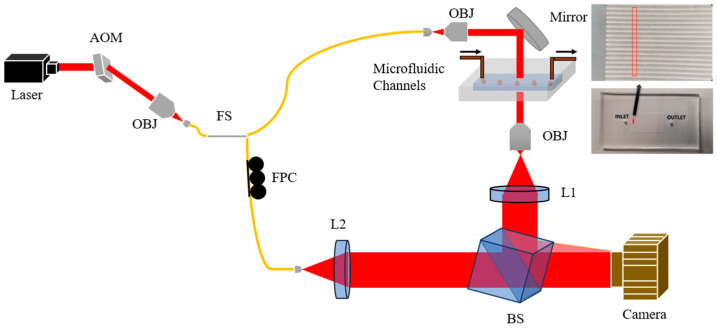
System Diagram. AOM, acousto-optic modulator. OBJ, objective. FS, 90:10 fiber splitter. FPC, fiber polarization controller. BS, beam splitter. L, lens.

**Figure 5 ijms-24-11885-f005:**
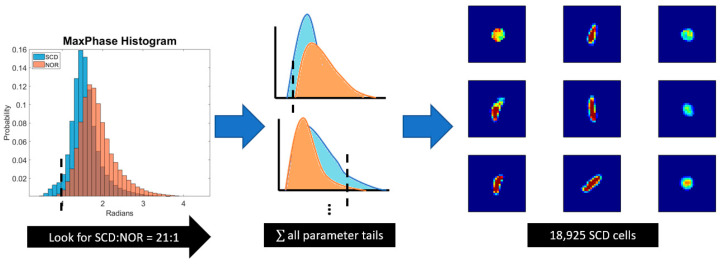
Refinement of training set data by selection of morphological parameter thresholds which are unique to the target cell population at a ratio of 21:1.

**Table 1 ijms-24-11885-t001:** Logistic regression-selected (LR-SEL) and logistic regression-all (LR-ALL) training accuracy. Averaged training sensitivity, specificity and accuracy of 5 trained logistic regression (LR) models, based on unrefined SCD datasets and refined SCD dataset.

Model	Sensitivity (%)	Specificity (%)	Accuracy (%)	Training Size (#)
LR-ALL	77.01 ± 0.21	75.14 ± 0.39	76.08 ± 0.20	200,000
LR-SEL	99.53 ± 0.17	99.03 ± 0.14	99.28 ± 0.09	37,850

**Table 2 ijms-24-11885-t002:** LR-ALL testing accuracy. LR model was trained using unrefined SCD dataset and unrefined normal dataset. Ground truth of the cells is labelled under column name ‘Type’.

Name	Type	Predicted: Normal (%)	Predicted: SCD (%)	#Cells
Sample A	Normal	99.64	0.36	483,975
Sample B	Normal	92.43	7.57	337,440
Sample C	Normal	99.21	0.79	283,996
Sample D	Normal	33.33	66.67	322,709
Sample E	Normal	56.75	43.25	133,990
Sample 3	SCD	59.16	40.84	93,060

**Table 3 ijms-24-11885-t003:** LR-SEL testing accuracy. LR model was trained using 18,925 selected SCD cell images and 18,925 normal cell images. Ground truth of the cells is labelled under column name ‘Type’.

Name	Type	Predicted: Normal (%)	Predicted: SCD (%)	#Cells
Sample A	Normal	98.66	1.34	483,975
Sample B	Normal	83.77	16.23	337,440
Sample C	Normal	96.77	3.23	283,996
Sample D	Normal	94.17	5.83	322,709
Sample E	Normal	97.46	2.54	133,990
Sample 3	SCD	70.74	29.26	93,060

**Table 4 ijms-24-11885-t004:** Convolutional neural network-all (CNN-ALL) testing accuracy. Convolutional neural network (CNN) performance table shows the percentage of cells predicted in different categories. The ground truth of the cells is labelled under column name ‘Type’. CNN model was trained using an unrefined dataset consisting of 100,000 SCD cell images and 100,000 normal cell images.

Name	Type	Predicted: Normal (%)	Predicted: SCD (%)	#Cells
Sample A	Normal	95.53	4.47	483,975
Sample B	Normal	92.40	7.6	337,440
Sample C	Normal	92.14	7.86	283,996
Sample D	Normal	46.93	53.07	322,709
Sample E	Normal	47.72	52.28	133,990
Sample 3	SCD	51.13	48.87	93,060

**Table 5 ijms-24-11885-t005:** Convolutional neural network-selected (CNN-SEL) testing accuracy. The ground truth of the cells is labelled under column name ‘Type’. CNN model was trained using refined dataset consisting of 18,925 SCD and 18,925 normal cell images.

Name	Type	Predicted: Normal (%)	Predicted: SCD (%)	#Cells
Sample A	Normal	99.27	0.73	483,975
Sample B	Normal	98.34	1.66	337,440
Sample C	Normal	98.96	1.04	283,996
Sample D	Normal	92.62	7.38	322,709
Sample E	Normal	94.37	5.63	133,990
Sample 3	SCD	74.04	25.96	93,060

**Table 6 ijms-24-11885-t006:** Current dataset patient demographics.

	Type	Age	Gender	BMI	#Cells
Sample 1	SCD	39	Male	52.69	236,032
Sample 2	SCD	39	Male	24.09	235,760
Sample 3	SCD	39	Male	25.22	93,060
Sample 4	Normal	53	Female	18.71	31,118
Sample 5	Normal	38	Male	32.61	192,409

**Table 7 ijms-24-11885-t007:** Archive dataset patient demographics.

	Type	Age	Gender
Sample A	Normal	35	Male
Sample B	Normal	20	Female
Sample C	Normal	32	Male
Sample D	Normal	56	Female
Sample E	Normal	65	Male

## Data Availability

All data used for this article are available upon request.

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
