# Peer review of "Biophysical Profiling of Sickle Cell Disease Using Holographic Cytometry and Deep Learning"

_ijms, 2023, doi:10.3390/ijms241511885_

Round 1

Reviewer 1 Report

Chen et al developed an approach to produce a highly accurate classification of samples from sickle cell disease patients and normal donors. Specific suggestions are as follows;

1. Abbreviations usage should be corrected. For instance, abbreviations should clearly be identified in Tables or SCD is abbreviated two times in the text.

2. Selection criteria of the samples should be mentioned in the results sections as it is very critical (not just reference the method section).

3. Discussion (i.e. future studies, what needs to be improved, limitations, etc) and literature support are very weak in the discussion part. 

Author Response

1. Abbreviations usage should be corrected. For instance, abbreviations should clearly be identified in Tables or SCD is abbreviated two times in the text.

Thank you for pointing this out. The authors have added clarification to the abbreviations in tables and graphs.

2. Selection criteria of the samples should be mentioned in the results sections as it is very critical (not just reference the method section).

We have added a paragraph in the results section explaining the selection criteria.

3.Discussion (i.e. future studies, what needs to be improved, limitations, etc) and literature support are very weak in the discussion part. 

Thank you to the reviewer for pointing out the weak points in the discussion, we have added two more paragraphs in this section to further expand on the future work needed to advance this work.

Reviewer 2 Report

In the manuscript entitled "Biophysical Profiling of Sickle Cell Disease using Holographic Cytometry and Deep Learning" Chen et al. provide evidence regarding the use of algorithms based on refined data to distinguish sickling cells from normal ones. The study is focused and to the point, while the results are clearly demonstrated.  

This Reviewer has some comments, as follow:

1.       Could the authors please provide the explanation for LR, CNN etc. abbreviations prior to their first use in the text? This reviewer believes it would be easier for the reader to follow.

2.       The authors’ findings are very informative and could prove to be useful in terms of categorizing the patients’ status regarding sickle cell disease, but this reviewer believes that the manuscript could be strengthened if a more detailed discussion regarding the authors’ findings is added.  At this point it would also be very interesting if a small piece of discussion was added focusing on the usefulness of such algorithms in other RBC disease settings as well (e.g. spherocytosis, beta thalassemia etc.).

3.       The authors report that a percentage window between, approximately, 8 and 25% represents a safe range to evaluate for SCD. What would be the accuracy of the model if a patient has more than 25% sickled cells? Could the authors also comment on the usefulness of their proposed model (especially the refined data-based one) in sickle cell trait individuals that have low incidents of sickling?

4.       The authors refer to the two groups under examination as SCD patients and normal patients. If this reviewer understood well, it would be preferable to replace “normal patients” with “normal subjects”.  

5.       In general, a slight proof-reading for minor mistakes throughout the manuscript might be needed.

In general, a slight proof-reading for minor mistakes throughout the manuscript might be needed.

Author Response

1. Could the authors please provide the explanation for LR, CNN etc. abbreviations prior to their first use in the text? This reviewer believes it would be easier for the reader to follow.

Thank you for pointing this out, we have corrected all abbreviations.

2. The authors’ findings are very informative and could prove to be useful in terms of categorizing the patients’ status regarding sickle cell disease, but this reviewer believes that the manuscript could be strengthened if a more detailed discussion regarding the authors’ findings is added.  At this point it would also be very interesting if a small piece of discussion was added focusing on the usefulness of such algorithms in other RBC disease settings as well (e.g. spherocytosis, beta thalassemia etc.).

Thank you for your suggestion, we have expanded the discussion to include the possibility of utilizing the presented methods in the paper for evaluating other types of RBC disease.

3. The authors report that a percentage window between, approximately, 8 and 25% represents a safe range to evaluate for SCD. What would be the accuracy of the model if a patient has more than 25% sickled cells? Could the authors also comment on the usefulness of their proposed model (especially the refined data-based one) in sickle cell trait individuals that have low incidents of sickling?

We thank the reviewer for providing these very interesting questions. We have added to the discussion section, to include our hypothesis on what we think percentage yields that fall outside of the percentage window (8% - 25%) would imply.

4. The authors refer to the two groups under examination as SCD patients and normal patients. If this reviewer understood well, it would be preferable to replace “normal patients” with “normal subjects”.  

Thank you for pointing this out, we have corrected the naming to “normal subjects”.

5. In general, a slight proof-reading for minor mistakes throughout the manuscript might be needed.

Thank you to the reviewer for your careful reading, we have proofread the manuscript and addressed the mistakes.

Round 2

Reviewer 1 Report

The authors clarified my concerns satisfactorily.  

Reviewer 2 Report

The authors have addressed my comments. I have nothing more to add.